# From Old Habits to New Routines—A Case Study of Food Waste Generation and Reduction in Four Swedish Schools

Christine Persson Osowski [1,2], Dariusz Osowski [3], Kristina Johansson [4], Niina Sundin [4], Christopher Malefors [4] and Mattias Eriksson [4,*]

1   Division of Public Health Sciences, School of Health, Care and Social Welfare, Mälardalen University, Box 883, 721 23 Vasteras, Sweden; christine.persson.osowski@mdu.se
2   Department of Food Studies, Nutrition and Dietetics, Uppsala University, Box 560, 751 22 Uppsala, Sweden
3   Department of Business, Society and Engineering, Mälardalen University, Box 883, 721 23 Vasteras, Sweden; dariusz.osowski@mdu.se
4   Department of Energy and Technology, Swedish University of Agricultural Sciences, Box 7032, 750 07 Uppsala, Sweden; Kristina.j-son@hotmail.com (K.J.); niina.sundin@slu.se (N.S.); christopher.malefors@slu.se (C.M.)
*   Correspondence: mattias.eriksson@slu.se; Tel.: +46-18671732

**Abstract:** Public food service organizations are large producers of food waste, which leads to greenhouse gas emissions and the waste of natural resources. The aim of the present article was to gain insight into reasons for food waste and possible solutions for lowering food waste in schools in Sweden. In order to do so, food waste quantification in school canteens in two Swedish municipalities and nine qualitative interviews with key actors were conducted. Both municipalities displayed a high degree of variation in food waste, but the common pattern was that serving waste constituted the largest fraction of food waste, followed by plate waste and storage waste, as well as a gradual decrease in food waste over time. Food waste was mainly a result of old, disadvantageous habits, such as overproduction due to forecasting difficulties, whereas new, better routines such as serving fewer options, better planning, and a less stressful environment are the key to lowering food waste. Because food waste varies from one case to the next, it becomes important to identify and measure the causes of food waste in each school in order to be able to establish tailor-made, conscious, and flexible food waste mitigation routines.

**Keywords:** food waste; school; quantification; interviews; canteen; routines



## 1. Introduction

About one third of all food intended for human consumption that is produced is estimated to be wasted yearly [1]. Sustainable Development Goal 12.3 aims to "by 2030, halve per capita global food waste at the retail and consumer levels and reduce food losses along production and supply chains, including post-harvest losses" [2]. Food service organizations, such as restaurants within the public sector, are large producers of food waste [3–6]. In Sweden, public food service generates about 75,000 tons of food waste yearly, and schools and pre-schools are the largest contributors to this waste with 51,000 tons [7]. According to the Swedish Environmental Protection Agency, Swedish schools have the potential to reduce their waste by 50% [8], which could lead to reduced greenhouse gas emissions, lower water footprint, saved natural resources [9], and reduced costs. However, because schools seldom engage in food redistribution activities, they are to a large extent limited to reduce waste at the source through improved resources management.

According to Swedish law, nutritious school lunches free of charge are to be served to all children of compulsory school age [10]. School lunches usually consist of one or more cooked dishes to choose from, served with vegetables, crisp bread with spread, and milk or water to drink. Besides the legal requirements, the Swedish Food Agency has issued

voluntary guidelines for school lunches, and among other factors the guidelines for school lunches inform how the environmental impact of the foods served may be lessened and gives advice on how food waste may be avoided [11]. This is in line with the general development in Sweden, which was one of the first countries to include environmental sustainability in its national dietary guidelines [12].

Several studies have looked at food waste quantification [13–17], but fewer studies have looked at reasons for food waste and possible solutions. Silvennoinen et al. conducted workshops with people working in the Finnish food service sector. Difficulties in predicting the amount of food to be produced and overproduction were put forward as the essential reasons for food waste, along with poorly liked recipes and attitudes and time limits among the staff. To reduce food waste, better planning and work management, education, and easy to use food waste quantification were put forward as important measures [16]. This is also in line with introducing a lean management approach in the food service sector to decrease food waste and reduce operational costs as suggested by Gładysz et al. [18]. Steen et al., who looked at risk factors for food waste in Swedish pre-schools and schools, found that plate waste increased with higher age, increasing portion sizes, and with larger dining halls, the latter possibly being due to increased noise levels and a stressful environment. Serving waste, i.e., food served that did not reach the plates of guests, and total waste increased with greater overproduction and was more profound in satellite kitchens, which receive pre-cooked food, than in production units, which prepare food onsite, because they do not have the possibility to cool and store left-overs. The authors concluded that a more accurate estimation of the daily number of children eating and how much food they consume is needed [9]. This is also one of many actions against food waste concluded by the Swedish Food Agency, who in 2020 released a handbook aimed at reducing food waste in public catering [19]. However, the actions listed in the handbook and their food waste reduction potential have not actually been explored and tested. It is therefore important to understand why food waste is being generated within this sector.

Swedish school lunches contribute to a substantial amount of food waste [7], making it an interesting case study. Research has shown that different kitchens have different causes of food waste and different opportunities to reduce it [20], and this calls for studies involving the staff and other key informants in finding the best solutions to reduce food waste [21]. According to Blondin et al., explanations of food waste and perceptions of food waste interact in informing mitigation strategies, and thus it is important to have a good understanding of both if food waste is to be reduced [22]. The present article does just that, and combines food waste quantification with qualitative interviews with school meal stakeholders. The aim of the present article was to gain insight into reasons for food waste and possible solutions for lowering food waste in schools in Sweden.

## 2. Materials and Methods

The present study is a mixed methods case study, including secondary analyses of interviews with key actors at the macro (national), meso (municipal), and micro (school kitchen) level focusing on food waste in schools, combined with direct quantification of food waste from these schools. By combining these data in the analysis, the study aimed to find reasons for food waste and possible food waste mitigation methods.

### 2.1. Selection of Participants

As common in qualitative studies, purposeful sampling, i.e., selecting information-rich cases for in-depth understanding based on the purpose of the study, was used to recruit participants [23]. A criterion for inclusion was that the material should be heteeogenous in its characteristics, striving for variation and diversity in data. In total, nine semi-structured interviews, with seven females and two males, were conducted at macro (national), meso (municipality) and micro (school kitchen) level.

At the macro level, three key actors at a national level were chosen based on their experience of working with food waste, with one informant specializing in public meals,

one representing a technical research institute, and one working as a project leader for a national food waste reduction program.

At the meso level, two managers working with public meals in central organizations at the municipal level were included. In order to represent different cases, these managers were chosen from two different municipalities: one smaller (henceforth referred to as municipality A) and one bigger (henceforth referred to as municipality B).

At the micro level, four schools were included from these municipalities, one from municipality A (henceforth referred to as A1) and three from municipality B (henceforth referred to as B1, B2 and B3). A criterion when selecting schools was that the schools should be heterogenous in their characteristics regarding, for instance, educational stage, size, and location. Two upper secondary schools, A1 and B1, with about 400 and 500 students, respectively, were chosen based on their large numbers of students and high levels of food waste. The kitchen in school A1 also produced food for two other satellite kitchens. Another upper secondary school, B2, with about 750 students, was chosen based on its sustainable and agricultural profile, and placement in the countryside, close to nature. This school also differed from the others in that the principal was responsible for the school food provision instead of the central organization at the municipality level, which was the case for the other schools. Finally, a large primary school, B3, with 700 students, was chosen in order to include younger pupils and a school with relatively low levels of food waste. Four interviews, one at each school, were conducted with school kitchen managers, and these interviews were combined with food waste quantification data from the schools, when available.

### 2.2. Food Waste Quantification

Food quantification data were collected from three schools (A1, B1, and B3, data not available from B2) before and after the interviews. All of the food waste quantification work was performed by the kitchen staff. Data was also collected from each municipality as a whole with measurements taken from 2012 to 2020. Municipality A, with around 3500 enrolled students [24], had 17 school kitchens serving meals to students aged 6 to 19 that provided data from spring 2014 to autumn 2020. Municipality B, with about 36,000 enrolled students [24], had a total of 72 school kitchens serving meals to the same age group as municipality B but their earliest food waste quantification efforts date from 2012 and their latest from 2020. The earlier quantifications encompassed one week each spring and autumn to gradually increase and cover more weeks over time; an overview of the development can be found in Table 1.

**Table 1.** Number of daily food waste quantification observations over time for municipality A and B.

| Year | Municipality A Observations (n) | Municipality B Observations (n) |
|---|---|---|
| 2012 | | 95 |
| 2013 | | 258 |
| 2014 | 74 | 273 |
| 2015 | 467 | 455 |
| 2016 | 509 | 430 |
| 2017 | 368 | 403 |
| 2018 | | 193 |
| 2019 | 138 | 894 |
| 2020 | 133 | 3977 |

The waste was split into three categories: Kitchen waste, i.e., waste that is produced in production kitchens, serving waste, i.e., waste that occurs during serving, never reaching the plates of guests, and plate waste, i.e., guests' unconsumed waste from plates. Kitchen waste can also be split into three subcategories: Storage waste, i.e., stored food items that become waste, preparation waste, i.e., waste from the preparation of food such as peels or bones, and safety margin waste, i.e., food produced that was never served and was not

saved for another meal. Because serving waste, plate waste, and storage waste are the largest fractions of waste, the results presented here will focus on these categories. These three fractions were also the common denominator between the analysed municipalities. However, municipality A had a much more ambitious quantification system, including several subcategories of kitchen, serving, and plate waste. A detailed mapping of the food quantification is displayed in Figure 1 for municipality A and in Figure 2 for municipality B, using the tree structure by Eriksson et al. [25]. The share of waste (%) of the different waste categories was also analysed as fractions of the total waste for the different kitchens and the municipalities.

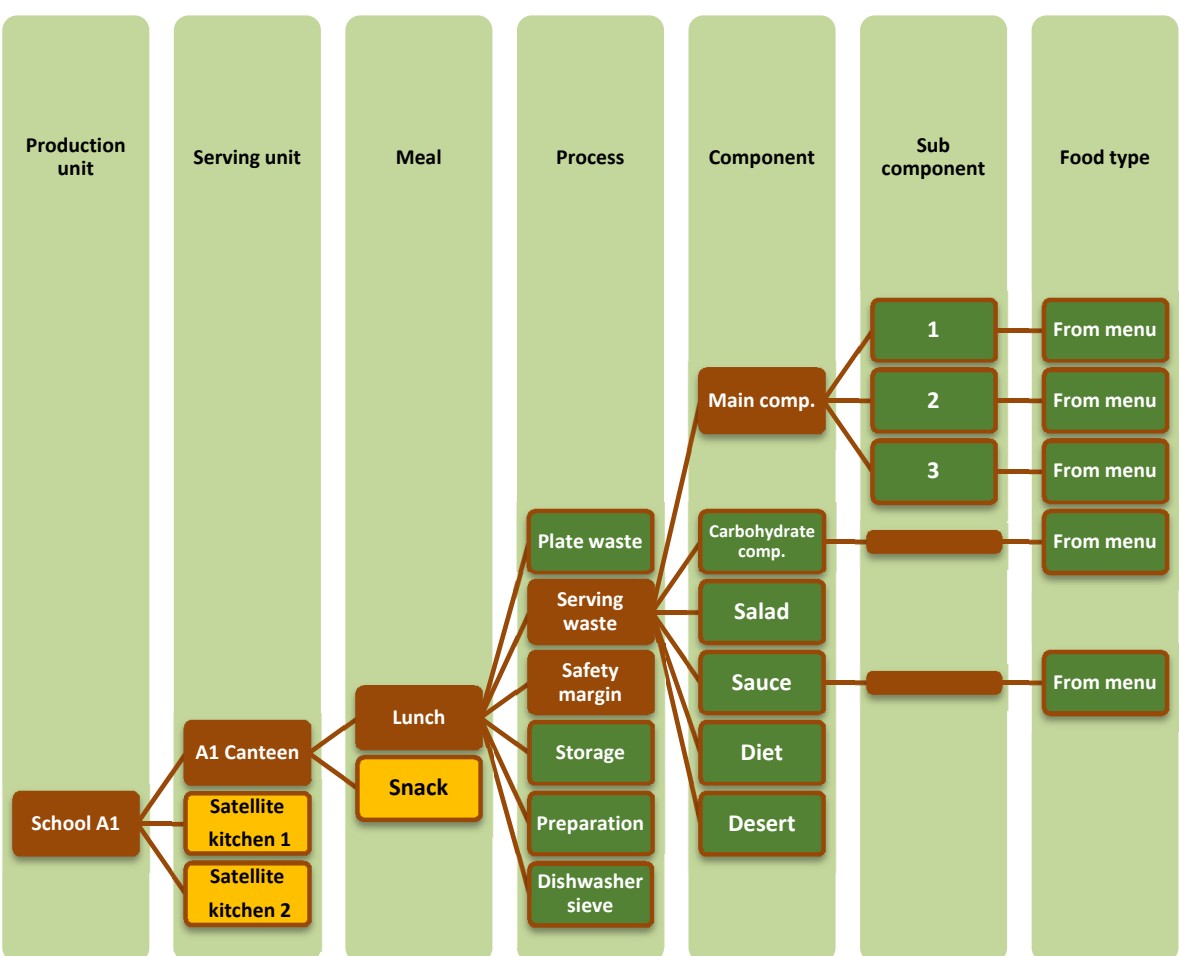

**Figure 1.** Production unit in secondary school A1 in municipality A which serves food both to a restaurant and to two satellite school canteens, with the safety margin aggregated with the serving waste when quantified for practical reasons. The production unit is displayed using the tree structure by Eriksson et al. [25] and the food types are not part of the quantification but can be connected through the menu in order to generate data on the type of food wasted. Brown areas in the diagram represent parent nodes to other nodes, orange nodes represent nodes that are present but not quantified, and green nodes represent the end-nodes that are quantified.

*2.3. Interviews*

The interviews were conducted in 2016 for a master's thesis. In order to map out key topics that might be of interest for the interviews, a pilot interview was conducted with a person representing a food waste reduction initiative. An interview guide (Appendix A) was then created and used during the actual interviews. The interviews were semi-structured [26] and the open-ended questions focused on reasons for food waste and what is being done to solve it. The informants' perceptions of the main problems with food

waste, causes of food waste and solutions to food waste were mapped out, and probing questions were used to get to the bottom of the problem and possible food waste mitigation solutions, ranging from micro to macro level. Nine semi-structured interviews were performed with key actors at the national, municipal, and kitchen level. All interviews were conducted in person, except for two interviews that were conducted over the telephone due to long distances. The interviews were audio recorded and transcribed verbatim, except for the two telephone interviews where notes were taken instead. The study was not in need of ethical vetting according to Swedish law [27]. The participants were informed about the present study in writing, and all gave their written consent to use the interview data for secondary analyses and publication in the present article. The informants were anonymized and therefore the municipalities and schools they represent were also coded.

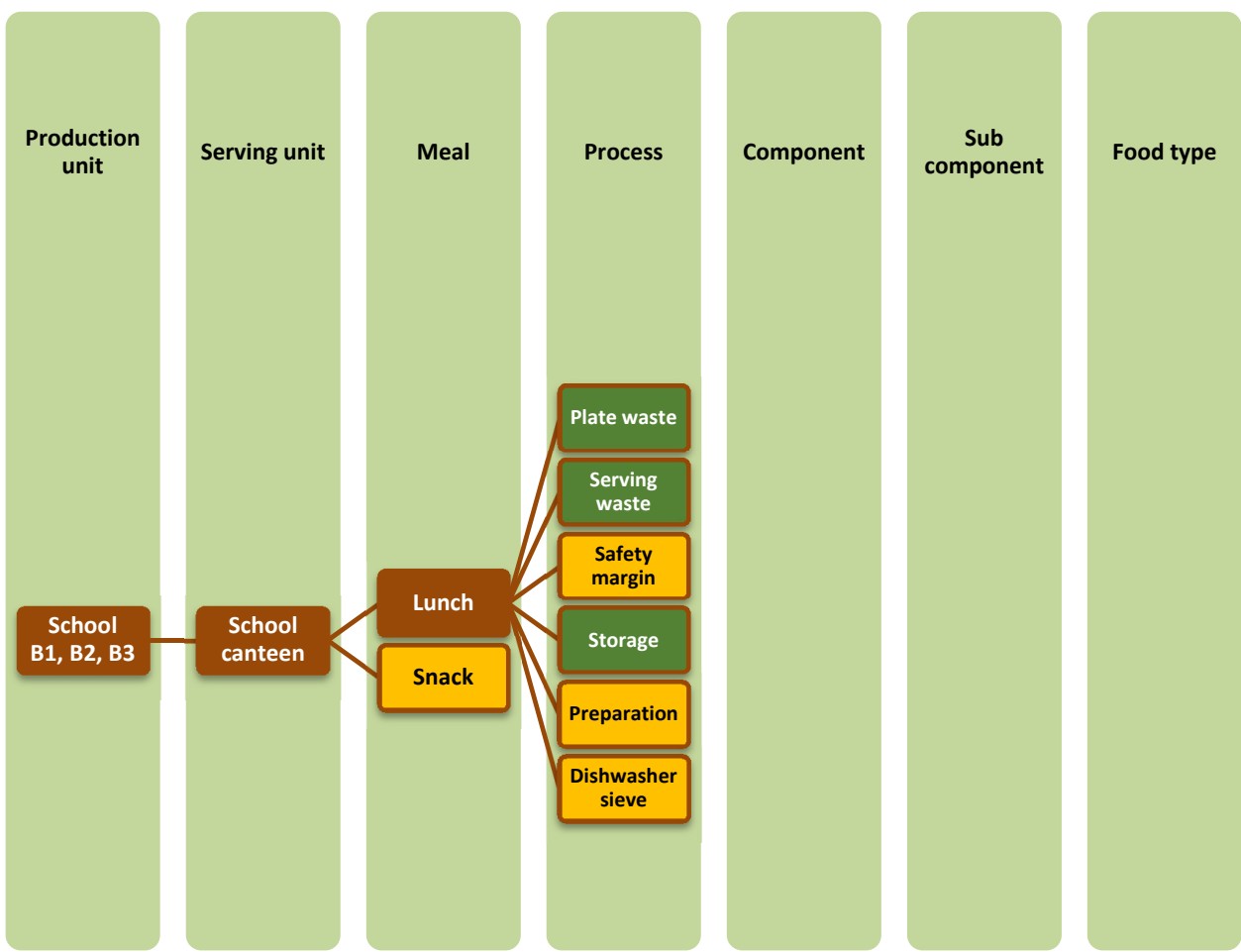

**Figure 2.** The school canteens B1 and B3 in municipality B. The school canteens are displayed using the tree structure by Eriksson et al. [25]. Brown areas in the diagram represent parent nodes to other nodes, orange nodes represent nodes that are present but not quantified, and green nodes represent the end-nodes that are quantified.

*2.4. Analysis*

The quantitative food waste data were compiled and visually inspected for possible errors. Days with missing data were removed from the dataset so that only days with complete observations, including the number of guests and at least one of the used categories of food waste with a recorded mass, were used. If zero guests were recorded in the dataset this was treated as a data gap. Key performance indicators, such as grams of food waste per guest, were then calculated for each school or municipality for the defined period including all available data, i.e., the calculated sum of all waste from

the different waste processes in kg was divided by the number of guests that attended i.e., $\sum \frac{Waste\ from\ the\ waste\ processes}{Number\ of\ portions\ served}$. All food waste quantification data were analysed by the framework proposed by Malefors et al. [28]. For comparative reasons, the median food waste (g) per guest from each municipality was also quantified on a yearly basis and aggregated for the whole quantification period. The yearly food waste levels are illustrated as boxplots with the individual contribution from each kitchen marked within the same illustration to arrive at an understanding of how the kitchens compare to other kitchens and municipalities.

As for the qualitative data, the transcribed interviews were imported into the qualitative data analysis software (QSR International NVivo Plus, 11, www.qsrinternational.com/nvivo-qualitative-data-analysis-software/home, (accessed on 16 November 2021)), and a thematic analysis [29] within a social constructionist framework was conducted. The social constructionist framework entailed that the accounts of the informants were seen as a result of the historical and cultural context [30]. Thus, the statements made by the informants should not be seen as a reflection of an ultimate truth, but rather as the informants' understanding of food waste. The analysis was secondary, meaning that the data were re-analysed for the purpose of the present study, looking specifically at why food was wasted and possible solutions to the problem. This process started with a reading of the data, analyzing the interviews inductively using codes that were kept close to the data and then merged into themes [29]. As a last step, the results were linked to a theory of organizational routines [31]. The theory of routines distinguishes two types of routines: performative (as an actual practice of a human on the individual level, i.e., routines in practice) and ostensive routines (as a commonly understood and expressed routine that can be communicated as a sort of a process to a broader audience, i.e., routines in principle) [32]. For the purpose of this paper, the first type of routine is referred to as habits, and the second is referred to as routines. Habits are usually unreflective and do not garner much attention because they aim to economize one's resources [33] and are often the result of path dependency (things are done in a certain way because they have been done that way for a long time). Routines, on the other hand, are more reflective and can necessitate attention and consciousness [34]. The final analysis resulted in two main themes—Food waste generation as a result of disadvantageous habits, and Better routines are the key to lower food waste—which are presented after the food waste quantification below.

## 3. Results

### 3.1. Food Waste Quantification

To achieve an understanding of what problems kitchens need to address, it is important to have an understanding of where actions to reduce food waste should be taken so that efforts are taken where the maximum potential for reduction lies. One way of establishing this is to look at the share of food waste in the different waste categories, which are shown in Figure 3.

Municipality A had a higher total level of food waste per served portion (65 g) compared to municipality B (53 g). In both municipalities, the largest fraction of food waste was serving waste, followed by plate waste and storage waste. All schools followed this pattern, with B3 being the only exception, as it did not report any storage waste and had a serving and plate waste that were similar to one another. B3 also differed from the other schools in that it reported a much smaller waste than the other schools (45 g per served portion as compared to 98 g in A1 and 64 g in B1, all years combined). Canteen B2 did not quantify its food waste and therefore these figures cannot be displayed. The variation of total food waste per served portion over time for the two municipalities is presented in Figure 4. Both municipalities displayed a large degree of variation.

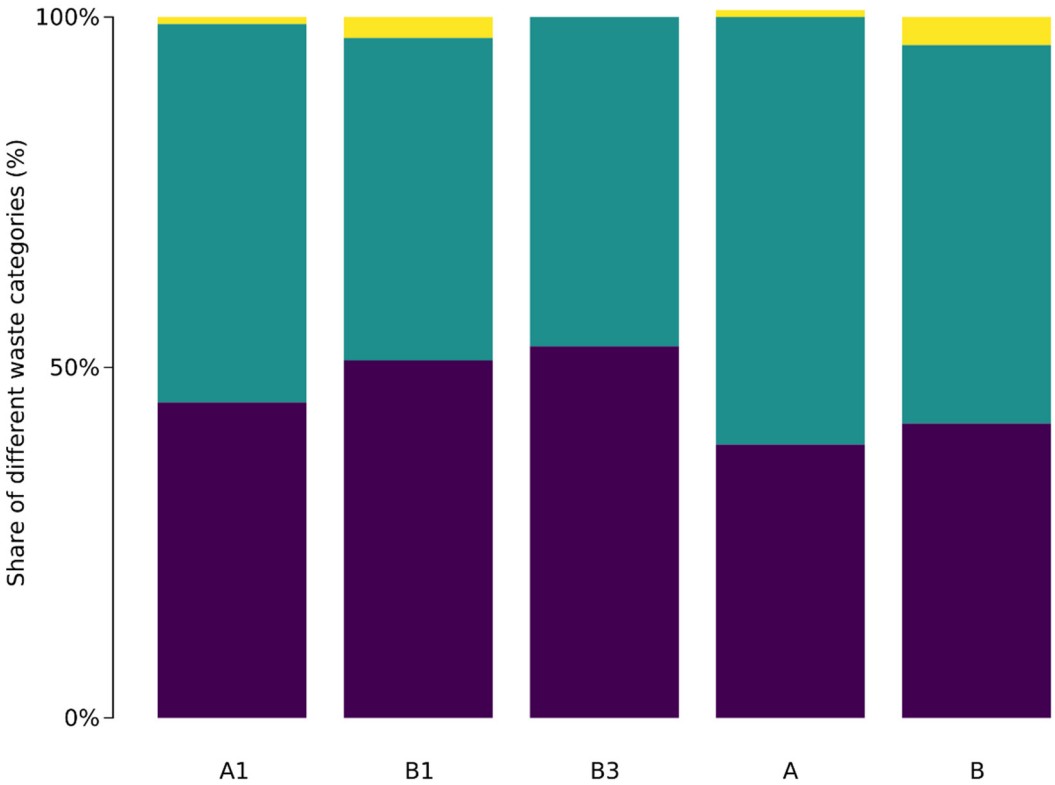

**Figure 3.** The share of different waste categories (plate waste ■, serving waste ■, and storage waste ■) for the schools and municipalities included in the study.

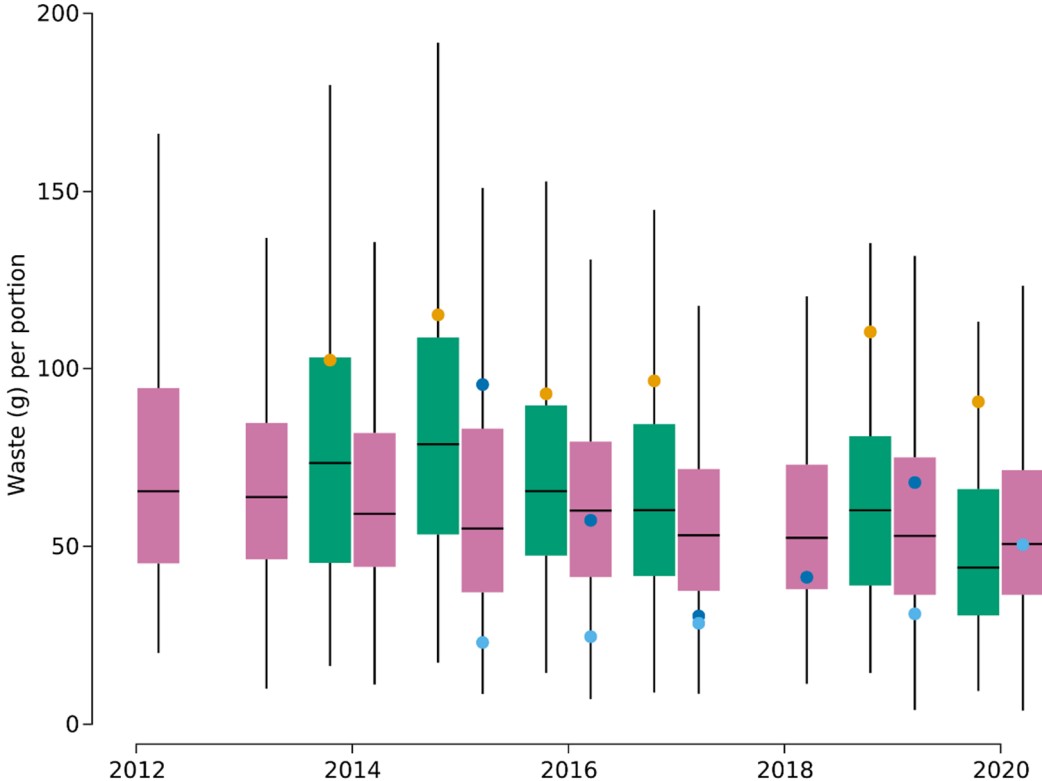

**Figure 4.** The variation in total food waste in grams per served portion over time for municipality A ■ and B ■. The school canteens median waste (g) per portion is also represented in the figure for schools A1 ●, B1 ●, and B3 ●.

### 3.2. Food Waste Generation as a Result of Disadvantageous Habits

The reasons for food waste generation were similar in the two municipalities. The biggest proportion of waste was serving waste, and this type of waste was produced due to overproduction and disadvantageous habits. One reason for this type of waste was difficulty in forecasting the amount of food to be produced, especially among the older pupils because they may choose to eat elsewhere instead of at school. Lack of communication between the school and the kitchen also contributed to forecasting difficulties, as the kitchen staff did not always know how many pupils were to be absent and thereby how many would be eating. Satellite kitchens, i.e., kitchens that receive warm food that has been prepared elsewhere, were mentioned as a hot spot for food waste because they may lack opportunities to cool down the food and thereby save it. However, although food hygiene was a reason mentioned by the kitchen staff for throwing away food, the informants at the national level mentioned that knowledge with regard to food hygiene needs to improve because food that is still good is often thrown away. The kitchen staff also put forward stress, lack of time, and fear of running out of food as reasons for overproduction and food waste:

> The biggest thing in this kitchen is the stress. Not enough staff, too stressful, you don't have time to think or plan, so you maybe overdo it and make some more because you are so scared that you will run out of food even when you know that 140 L would have been enough. But you don't have the time. At least not the way we have had it. Kitchen manager, school A1.

On the other hand, another kitchen manager spoke of finding it difficult to get the staff to change their old habits and to accept new routines:

> You want it to be simple and you are used to doing things in a certain way. And you have a difficult time accepting a new way, "I have always done it this way", well, yes, you have always made two tin plates too much, so maybe it is time to make two tin plates less. Kitchen manager, school B1.

The second largest proportion of waste was plate waste. The informants put forward that one contributing factor to plate waste could be that school meals have a low status and that the children do not cherish the food because it is served without cost. On the other hand, it was commonly mentioned that popular dishes are being wasted to a larger extent because the pupils habitually fill up their plates with a large amount of food that they are unable to finish. Scheduling issues and not giving the students enough time to eat was another reason for plate waste:

> A lot of people who throw away the food feel that they don't have time to serve themselves because the lines are too long and there are many people arriving at the same time. So then you serve yourself with more food and might throw it away because you know that you won't have time to serve yourself again. Or you are unable to finish your meal because you are not given enough time to eat. Or your friends leave, and you don't have enough time. Manager, municipality B.

Storage waste only constituted a small fraction of the total waste and was discussed to a lesser extent in the interviews. Causes of storage waste could be habits of buying too large quantities, buying the wrong products, receiving products with a short use-by date or of poor quality, wrong deliveries, or simply not using up the products that had been previously bought.

### 3.3. Better Routines Are the Key to Lower Food Waste

The measures that had been taken to reduce food waste were similar in the two municipalities. Having functioning routines in measuring the food waste, evaluating figures and measures being taken, and forecasting how much to produce were the main keys to success. The biggest proportion of waste, i.e., serving waste, was the type of waste that the informants at the national level thought should be targeted to a greater extent,

and it was also the type of waste that the staff thought they had the biggest chance of influencing. Here, forecasting the amount to produce was important. The kitchen manager at the school with the lowest amount of waste described their working routines with regard to forecasting in the following way:

> And as I said, we always take notes. And we have a schedule that we cook the food here and a schedule so we know how many pupils are still to eat, so they cross out the ones who have already eaten. Then we only have to look at the schedule how many that remain and are to come, so then you do not have to cook so much food and throw it away. Kitchen manager, school B3.

The teachers were seen as important in influencing the children in lowering plate waste. One way that this could be done was to apply so-called "pedagogic lunches" routines, with the teachers eating with the children and teaching them about food and the environment while eating. With teachers being present in the canteens, there was a possibility to influence how much the children put on their plates and thereby lowering the plate waste, for instance, by saying that they could serve themselves more than once. However, how this worked differed, for instance, due to the different level of involvement of the teachers and the age of the children, and pedagogic lunches are mainly applied in lower age groups. Another routine mentioned could be to teach the pupils during regular school hours, for instance, during home and consumer studies, in order to make them more aware of food's impact on the environment. Moreover, one of the informants at the national level talked about the importance of routine in informing the children in a positive way by focusing on enjoyment of food and getting the children to eat rather than informing them that they should throw away less. The importance of cooperating with different actors in order to lower food waste, especially the teachers, was brought up by the manager in municipality A:

> But we need to cooperate with teachers, schools, and politicians so that the politicians can put pressure on all actors in the organization. And especially the teachers to make sure that the pupils do not serve themselves more food than they are able to finish. Manager, municipality A.

Serving fewer dishes each day had proven to result in a lower amount of waste. Municipality A had measured the effects of serving one dish less, i.e., two dishes instead of three daily, and this had resulted in both lower serving and plate waste. Having flexible menus, where the leftovers could be served another day, had also proven to be successful in both municipalities. One of the schools had recently switched to a routine of serving the food as a buffet, presenting different components rather than set dishes, thereby allowing for flexibility. Other measures that had been taken at the various schools included routines of serving less food at a time, rewarding the children for lowering plate waste, for instance, by serving ice-cream, placing small spoons by the food for the pupils to try the food before they serve themselves, educating the staff, recruiting knowledgable staff from the private sector, or arranging lectures with star chefs. While the informants at the national level saw food policies as a potential way to lower food waste, the kitchen staff preferred local solutions to solving the food waste problem. The most important reasons for food waste and waste mitigation strategies mentioned by the informants are summarized in Table 2.

**Table 2.** The most important reasons for food waste and waste mitigation strategies mentioned by the informants.

| Reason for Food Waste | Mitigation Strategy |
| --- | --- |
| Serving waste | Measuring food waste |
| | Forecasting |
| | Cooperation and communication |
| | A less stressful work environment |
| | Education and knowledgeable staff |
| | Flexible menu and work routines |

**Table 2.** *Cont.*

| Reason for Food Waste | Mitigation Strategy |
|---|---|
| Plate waste | Education/Pedagogic lunches<br>Enough time to eat<br>Possibility to try food before serving themselves<br>Rewards |
| Storage waste | Better purchasing routines |

## 4. Discussion

Food waste quantification showed a variation between the municipalities, between schools, and within schools over time. One school (B3) had a substantially lower total amount of food waste than the others. One possible explanation is that the pupils were younger, as previous studies have shown that plate waste increases with the increasing age of pupils [9]. Further, the kitchen staff of this particular school had a forecasting routine that may have contributed to their lower amount of serving waste, as seen in previous studies [35].

In both municipalities, the largest fraction of food waste was serving waste, followed by plate waste, whereas storage waste only contributed to low levels of waste. The serving waste was mainly generated due to overproduction. Goonan et al., who looked at hospital food service, also found that overproduction is the main cause of food waste [4]. Overproduction mainly seems to be a result of poor prediction, and better forecasting routines could be a solution to this [16,35]. The possible solutions raised by Silvennoinen et al., i.e., better planning and work management, education, and easy to use food waste quantification [16], are solutions that are in line also with the solutions mentioned by the informants in the present study. For example, forecasting was considered more challenging by the staff that served older pupils due to older pupils often choosing to eat elsewhere. Previous studies have shown that skipping school lunch becomes more common with age [36,37]. In addition, to be able to forecast accurately, communication between the school and the kitchen must work so that the staff knows how many pupils are expected to be present each day, an area that was identified for improvement. In order to aid forecasting, a food waste tracking system based on the Internet of Things could be of help, as it points out the reasons for food waste. A British company producing ready meals was able to reduce its food waste by about 60% by using this system and educating the staff. Some of the new routines implemented by the company included using trim waste from vegetables and chicken in soups, making sure less ingredients expired, correcting mis-orders, less spoilage, and preventive actions to diminish equipment failure [38].

Attitudes, such as the need to ensure that the quantity of food is enough, and old, often unreflected habits of the staff may also negatively influence the amount of waste. In the present study, having good work routines [32] and educating staff were mentioned as possible solutions. The staff also put forward that a work practice with less stress is necessary to reduce food waste. Depending on the type of kitchen, the causes of food waste differed. Satellite kitchens, i.e., kitchens that received warm food that had been prepared elsewhere, were mentioned by the informants as a large contributor to serving waste because they lack opportunities to cool down the food and thus to save it, which is in line with previous research [9]. Some production units, on the other hand, were mentioned by the informants as having routines to use leftovers when flexible menus are applied, which was identified as a corrective strategy in previous literature when halting the conversion of surplus into waste by reintroducing it back into the system [39]. Thus, different types of kitchen have different opportunities to solve the problem of serving waste.

The second largest fraction of food waste was plate waste. Just as with serving waste, age seems to be an important factor as Steen et al. found that plate waste increased with higher age [9]. Education and routines involving pedagogic lunches were mentioned by the informants as possible solutions to reducing plate waste. Steen et al. also pointed out

large dining halls as a reason for plate waste [9], so dividing the dining hall into smaller sections and focusing on creating a calmer meal environment could possibly contribute to lower plate waste, but this calls for further studies. The informants also mentioned school meals having low status as a reason for plate waste. Previous studies have also shown that people have negative preconceived ideas towards institutional meals [40,41]. The informants also mentioned that popular dishes are being wasted to a larger extent because the pupils habitually fill up their plates with a large amount of food that they are unable to finish; however, this was assessed by Eriksson et al. who found the difference in terms of food waste generation between popular dishes and unpopular dishes to be very small [42]. Scheduling issues and not giving the students enough time to eat was another reason for plate waste, which has been noted previously [43].

The present study has some limitations. The interview-related material used for the study was limited because only nine interviews were conducted, so it cannot be concluded that saturation was achieved [44]. Moreover, the qualitative nature of the data does not allow for generalisations but should be interpreted as the informants' understanding of food waste generation and reduction. In future studies it would be beneficial to conduct large scale surveys to allow for generalisations. However, the combination of interview data and statistics from food waste quantification strengthens the results. Further, the qualitative data were collected a few years prior to the publication of the present article, and it is therefore possible that the situation looks different now. Lastly, the food waste quantifications were conducted periodically instead of continuously, covering data collection periods between one and four weeks, (except for the autumn of 2020 for municipality B when they introduced food waste quantification as a daily routine), which could cause the data to miss some of the seasonal variations in food waste generation. However, quantifications were conducted each fall and spring over a few years' time period, thus minimizing the aforementioned limitation. Another limitation is that kitchen B2 never quantified their levels of food waste. This highlights the need for a more data-driven approach to know what efforts should be in focus. Both municipalities have, over time, gradually decreased their levels of food waste, which shows that the systematic work has had some effects. Food waste quantification in both organizations enables them to investigate what types of measures against food waste have an effect and which do not. In 2020 the Swedish Food Agency released a handbook for public catering canteens which suggests measures to reduce food waste, and a key element in the handbook is to have a quantification in place to evaluate whether the new routines have the desired effect [19].

As previously mentioned, different kitchens have different causes of food waste and different opportunities to reduce it [20]. Filimonau and Coteau [21] argue that from an organizational and stakeholder perspective, reduction of food waste requires an understanding of its importance by people who are familiar with the issue and who are also capable of making decisions on behalf of the kitchen operation. In the case of food waste, managers or staff working in the kitchen are such people, because not only do they define what food to order and cook and how to serve it, but they are also in charge of decision-making on the floor. Therefore, managers and staff need to reflect upon their knowledge and experience of dealing with this issue, both in terms of scale and scope, as well as the underlying causes of food waste generation. Here the combination of quantitative and qualitative data is fundamental to obtaining a deeper understanding of what actions to take to reduce food waste in the local setting.

## 5. Conclusions

This study showed that food waste is mainly a result of old, disadvantageous habits, whereas new, better routines such as serving fewer options, better planning and a less stressful environment are the key to reducing food waste. Because food waste varies from one case to another, it is important to quantify and identify the causes of food waste in each school in order to be able to establish tailor-made, conscious, and flexible food waste mitigation routines. Both municipalities have gradually decreased their level of food waste

over time, which indicates that their systematic work against food waste on all levels within the organizations has had an effect. In order to achieve a more sustainable public catering that generates less food waste, policymakers should therefore focus on policies that provide the right environment for these systematic improvements, rather than focusing on predefined actions.

**Author Contributions:** Conceptualization, D.O.; methodology, C.P.O., M.E. and K.J.; software, M.E. and C.P.O.; validation, all authors; formal analysis, C.P.O.; investigation, K.J.; resources, M.E.; data curation, C.P.O., M.E. and K.J.; writing—original draft preparation, C.P.O.; writing—review and editing, all authors; visualization, M.E.; supervision, M.E.; project administration, M.E. All authors have read and agreed to the published version of the manuscript.

**Funding:** This research received no external funding.

**Informed Consent Statement:** Informed consent was obtained from all subjects involved in the study.

**Data Availability Statement:** The data presented in this study are available on request from the corresponding author. The data are not publicly available due to ethical reasons.

**Acknowledgments:** The authors would like to thank all participants for taking part in the study.

**Conflicts of Interest:** The authors declare no conflict of interest.

## Appendix A

*Interview Guide*

- Why do you consider food waste to be a problem?
- If and how do you work with food waste? What focus? Individual level or structural level with more policies and rules?
- What do you consider to be most important regarding this issue?
- What's the problem with food waste?
- What are the reasons?
- What are the reasons for these reasons?
- Why is it a reason? Why isn't anything being done to prevent food waste?
- What's your opinion about political instruments and policies to reduce food waste?

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
