# Peer review of "From Old Habits to New Routines—A Case Study of Food Waste Generation and Reduction in Four Swedish Schools"

_resources, doi:10.3390/resources11010005_

Round 1

Reviewer 1 Report

In my opinion the authors' corrections have greatly increased the scientific soundness of the manuscript. They responded to each of my comments and suggestions. However, I have a few more:

I stand by my words about the choice of methodology - interviews do not provide a good insight into the discussed problem.

I believe that it would be beneficial for the article to add information on the number of meals served in individual municipalities; due to the area of ​​the research, it is important information that affects the amount of food waste generated.

Adding the equation to the data collected in lines 209-216 would increase the clarity of the information presented.

I consider the information from lines 258-262 to be incorrect from the scientific point of view, and the conclusions drawn here are supported by conjectures ("Due to a lack of data, figures from B2 are not displayed, but the kitchen manager claimed that their level of waste was about 10 kg per day and that this was mainly plate waste, as they used the leftovers. This would yield 13 g per portion if all of the enrolled students were present and the waste estimate from the manager is correct").

The article lacks clear conclusions that indicate the ways of solving the problem of food waste production in the considered municipalities. The manuscript highlights the problem, but doesn't offer specific actions that should be taken.

Reviewer 2 Report

Abstract – Results from this research are vaguely explained. Please ensure you briefly explain them.

Intext citations needs to follow the MDPI style. For example, 1 to be replaced by [1].

Introduction section – Some of the information provided has been repeated. For instance, paragraph 2 has a sentence” According to Swedish law, nutritious school lunches free of charge are to be served….”, which has been repeated in paragraph 3 “Sweden is one of few countries in the world where school lunches are served free of”. There are other similar sentences, please go through the whole Introduction section and correct it.

Defining purposive sampling would be beneficial.

Following paper’s Section 5.1 could be useful for this paper: The digitisation of food manufacturing to reduce waste – Case study of a ready meal factory. This paper has some other reasons for food waste generation which have similarities to food catering facilities.

High number of self-citations - Malefors & Eriksson – 9, Johansson – 1, Persson Osowski -7, Sundin – 1. Need to restrict these below 5 between all authors.

I have come across similar researches to yours but how is your research different to those needs explaining.

Round 2

Reviewer 2 Report

No comments